# The Influence of Short-Term Heavy Rainfall on Hydraulic Characteristics and Rill Formation in the Yuanmou Dry-Hot Valley

**DOI:** 10.3390/ijerph192215232

**Published:** 2022-11-18

**Authors:** Jun Luo, Xueyang Ma, Lei Wang, Bin Zhang, Xiao Yang, Tianxiang Yue

**Affiliations:** 1School of Geographical Sciences, China West Normal University, Nanchong 637009, China; 2Institute of Geographical Sciences and Natural Resources Research, Chinese Academy of Sciences, Beijing 100101, China; 3Sichuan Provincial Engineering Laboratory of Monitoring and Control for Soil Erosion of Dry Valleys, China West Normal University, Nanchong 637009, China; 4College of Environmental Science and Engineering, China West Normal University, Nanchong 637009, China

**Keywords:** washout duration, rill development, flow hydraulic characteristics, runoff development, short-term heavy rainfall

## Abstract

Rill erosion is one of the major environmental problems in the world; it is an important factor with regard to land degradation and has a serious impact on production and daily life in the region. The widely distributed Yuanmou group stratum promotes the development of rill erosion, whereby the strong time-concentrated rainfall and the alternating arid-humid climate prepare the ground for the development of rills in soils. Therefore, a study of the processes of slope rill erosion was carried out, and a gravel-soil slope in the Yuanmou dry-hot valley was chosen to simulate short-term heavy rainfall (25 mm/h) (No. 1 plot) and moderate rainfall (15 mm/h) (No. 2 plot), to study the erosion processes of soil and the dynamic characteristics of runoff involved in erosion. The study results showed that the width of runoff was significantly different between the two plots, while the depth of runoff was not significantly different. During the rill formation process, the width of the two plots first decreased and then increased with increasing washout duration, while its depth did not change significantly. Flow was the key factor in determining the hydraulic characteristics of runoff, and it had a significant or extremely significant positive correlation with hydraulic characteristics parameters, except in the case of *Fr* (Froude number) (*r* = 0.039). The total sediment content (CS) of plot No. 1 (0.158 g/cm^3^) was significantly different from that of plot No. 2 (0.153 g/cm^3^), and both CSs in the two plots decreased with increasing washout duration. The CS had an extremely significant negative correlation with τ (runoff shear force) (*r* = −0.863 **) and *DW-f* (Darcy-Weisbach drag coefficient) (*r* = −0.863 **) and a significant negative correlation with Re (Reynolds number) (*r* = −0.735 *) in the short-term heavy rainfall experiment, while the CS had a significant positive correlation with *V* (velocity) (*r* = 0.814 *), *R* (hydraulic radius) (*r* = 0.811 *) and *P* (unit stream power) (*r* = 0.811 *) in the moderate rainfall experiment. The results of this study will help guide further examination of the processes involved in the dynamic mechanisms of rill erosion on slopes under short-term heavy rainfall conditions.

## 1. Introduction

Soil erosion is a serious threat to soil and water resources [1,2] and is a severe environmental problem in terms of human survival and development [3]. The processes of soil erosion are complex, including the interactions between water flow and soil stripping, sediment transport, and deposition [4,5]. Soil erosion occurs in many ways; rill erosion represents one of the most important erosion methods, its sediment yield accounting for approximately 50% of the total sediment transport [6,7,8]. The main driving force for rill erosion and runoff sediment transport is flow, which is the key to linking rill erosion and channel erosion [9], and the velocity of runoff flow is related to the complexity of rill microtopography [10,11]. Rainfall intensity is significantly positively correlated with runoff and erosion [12]; with increasing erosion intensity and energy, the maximum erosion and sediment yield of slope transitions increases from the rill erosion zone to the shallow trench erosion zone [13,14]. In addition, there is a close relationship between the runoff shear force (*τ*) and the sediment transport rate under the same underlying surface conditions [15], and the properties of soil and the hydraulic characteristics of runoff are the decisive factors of rill erosion, while the sediment yield of a rill depends on the hydraulic characteristics of runoff [16,17,18]. Therefore, the study of the hydraulic characteristics of flow involved in rill erosion is important for exploring the mechanics of rill erosion and its dynamic processes [5].

The depth and width of runoff of rills are far less than those of gullies and rivers, which causes the hydraulic characteristics of rills to have a significant difference from those of gullies and rivers [19,20,21]. The hydraulic characteristics of rills are affected by slope length, slope height, rainfall characteristics, and soil properties, but the differences in natural and critical conditions for the development of rills are not fixed [22,23,24]. As the basic parameter used to characterize the hydrodynamic characteristics of runoff, the flow pattern is the basis for analyzing the properties of slope flow, runoff erosion, and sediment transport, which are closely related to the conditions of the slope [25,26,27]. The gradient and shapes of slopes, being the most important factors affecting rainfall runoff and soil erosion, have a direct impact on the patterns of runoff flow and the rill density [28,29]. Conversely, the velocity and depth of flow are controlled by runoff, and the magnitude of runoff affects the flow state directly [23,24].

To overcome the difficulties inherent in conducting experiments during the natural rain cycle, controlling for the rainfall time period and poor soil texture, most studies on the hydraulic characteristics of slope runoff have examined discharge scouring or carried out simulated rainfall tests with small volumes of simulated rainfall (5–25 L·min^−1^) [30,31,32]. However, the experiments were carried out in different regions and soil types, and they focused on loess areas [10,33,34], black soil areas [35,36,37,38,39], red soil areas [40,41,42], and purple soil areas [43,44]. The slope-gully system of the Loess Plateau is the main source of sediment in the basin, and it is also the basic unit for controlling soil erosion and for restoring and rebuilding the ecological environment [45,46,47] For many years, researchers have studied the effects of erosion and sediment production in the slope-gully system, and have conducted in-depth research on the temporal and spatial changes of the erosion mode, the slope-gully system formation [46,48], the sediment source, and the ectopic erosion effect of water and sand from above [49,50]. Black soil erosion mainly refers to the black soil region in northeastern China; the black soil loses the protection of vegetation due to the high-intensity reclamation and unreasonable demands of farming by humans [46] and because rainfall intensity is high in summer [51], both of which accelerate soil erosion and then aggravate the loss of water, fertilizer, and soil, further reducing land productivity in the black soil region [52,53]. Meanwhile, accelerated soil erosion would aggravate floods and affect the comprehensive development and utilization of water and soil resources [51,54,55,56,57], etc. Research into purple soil mainly focuses on the soil’s physicochemical properties and soil erosion, and rainfall has a great influence on soil properties. By studying the changes in the degree of soil erosion under rainfall conditions, we can fully understand the process of soil erosion in purple soil [58], and provide a theoretical reference and scientific basis for soil and water conservation in the purple soil area. External factors will have a certain impact on purple soil erosion, such as land use, slope length, and farming methods [9,59,60], which will lead to changes in the purple soil’s properties and soil quality.

However, there have been few studies on the dry-hot valley of the Jinsha River, where soil erosion is severe. The geographic, environmental, and climatic conditions of the Yuanmou dry-hot valley have caused the regional topography to be extremely fragmented [61,62]; rills have developed and soil erosion has seriously threatened the survival and economic development of the region’s residents [63,64,65]. In turn, environmentally considerate construction in the Yuanmou dry-hot valley is the key to harnessing the Yangtze River Basin and protecting the Three Gorges Dam. Moreover, the Yuanmou dry-hot valley has less annual rainfall and longer dry seasons but more concentrated rainy seasons than the dry valleys of the Min River, Yalong River, and Nu River in southwestern China [66]. The long dry season leads to a loose surface of dry red soil in the region, which is easily washed away by the runoff caused by short-term heavy rainfall, resulting in increased soil erosion. Therefore, the Yuanmou dry-hot valley was chosen as the research area, and a high-flow erosion test was carried out in situ in the field. Using the real-time monitoring of high-flow runoff erosion, the morphological characteristics of rill development and the hydraulic characteristics of rill runoff under different treatments were analyzed systematically to reveal the characteristics of the runoff hydraulics involved in the process of rill erosion and thereby provide the necessary theoretical basis for the rational planning of slope soil and water conservation.

## 2. Materials and Methods

### 2.1. Study Area

The Yuanmou dry-hot valley is located in Yuanmou County (25°23′–26°06′ N, 101°35′–102°06′ E), Yunnan Province, and is part of the Jinsha River dry-hot valley (Figure 1). The climate is hot and dry due to the Foehn effect, resulting in an average annual precipitation in the valley of approximately 630 mm; the average annual evaporation is five times the precipitation [67]. Precipitation during the rainy season (May-October) accounts for 91% of the annual rainfall [68]. The soil types in the valley include dry red soil and vertisol. In some areas, much topsoil has been eroded, and the vertisol has been exposed (Figure 1). The clay content of the soil is high, and erosion resistance is considerable. The vegetation in the valley is dominated by grasses, although shrubs and trees are sporadically distributed, with a low forest coverage rate of only 3.4–6.3%. These factors, coupled with poor climatic conditions, have caused serious rill and gully erosion to occur in the valley.

### 2.2. Experimental Design

The study site is located on the hillslope of Julin Village (25°50′57.47″ N, 101°49′50.32″ E), Huanguayuan Town, Yuanmou County (Figure 1). The farmland in the study area is fragmented and the slope is mostly a gentle slope (<15°). In the study area, linear rills formed on farmland in rainy seasons, which caused severe soil erosion. The short-term heavy rainfall washout experiment was carried out in the field, where there were two test plots facing westward, each 20 m × 5 m, and the slope was approximately 8° (Figure 2). The washout water was supplied by a rainfall reservoir in a water supply tank on the upper part of the two test plots, with a collection tank on the lower part. The boundary of the two test plots was made of mortar and concrete to ensure that there was no water leakage. The soil of the plots was divided into upper and lower layers. The upper layer consisted of the typical dry red soil of the local area, and samples of this soil were collected within 20 m of the test plot. The lower layer consisted of vertisol; soil samples were collected from nearby areas (<1 km). Dry red soil is also known as Jianyu dry-moist iron-rich soil, in terms of systematic classification. Due to weak weathering and eluviation, dry red soil belongs to sandy soil, which is mainly characterized by high sand content and good permeability. Vertisol has strong swelling, shrinking, and disturbance characteristics, which belong to soil with high clay content and poor permeability. Compared with dry red soil, vertisol has a higher pH value and cation exchange capacity. The depth of both soil layers was 0.5 m; the plot materials, such as stones or clods, were not cleaned and retained their original soil-gravel composition. However, the plants in the test plot were removed before the test and the slope was flattened. A waiting period of approximately 30 days followed so that the test slope was close to the condition of a real slope.

### 2.3. Data Calculation

The velocity, depth, width, and water temperature data of the runoff in each section were measured nine times, and the average data were used to calculate the parameters of the hydraulic characteristics, as described below [20].

Hydraulic radius (*R*)

The section shape of the test plot was rectangular, and the formula for calculating the hydraulic radius is shown in Equation (2):(1)$$R=\frac{hb}{b+2h}$$
where *b* is the width of the runoff and *h* is the depth of the runoff in cm.

Reynolds number (*Re*)

In fluid mechanics, *Re* is the ratio of the inertial force to the viscous force of a fluid and is used to predict whether the flow will be laminar or turbulent; *Re* can be calculated as follows in Equation (3).
(2)$$Re=\frac{VR}{v}$$
where *V* (m/s) is the velocity of runoff at a cross-section and *v* (m^2^/s) (Equation (4)) is the motion viscous coefficient of the slope water flow, which is determined by the water temperature (*t*):(3)$$v=\frac{0.01775}{1+0.337t+0.00022t^{2}}$$

Froude number (*Fr*)

The Froude number is a dimensionless parameter that characterizes the relative inertia of a fluid and the relative magnitude of gravity, representing the ratio of inertial force to heavy force level. When *Fr* < 1, the water flow pattern is slow, and when *Fr* ≥ 1, the water flow state is rapid.
(4)$$Fr=\frac{v}{\sqrt{gh}}$$
where *g* is the gravitational acceleration (9.8 m/s^2^).

Darcy-Weisbach drag coefficient (*DW*-*f*)

The Darcy-Weisbach drag coefficient represents the resistance of soil to water flow. The larger the value of *f* is, the more serious the soil erosion.
(5)$$f=\frac{8gRJ}{V^{2}}$$
where *J* represents the hydraulic gradient and is the sine of the slope. The slope of the test plot was 8°, and *J* = 0.139173101.

Runoff shear force (*τ*)

Runoff shear force can be expressed as follows:(6)τ=ρghS
where *ρ*, *g*, *h* and *S* are the density of water (1000 kg/m^3^), acceleration due to gravity, the runoff depth, and the test plot slope, respectively.

Unit stream power (*P*)

The unit stream power is given by:(7)P=VJ
where *V* (m/s) is the velocity of runoff at a cross-section, (m/s) is the velocity of runoff at a cross-section.

Cross-section unit energy (*ε*)

The cross-section unit energy of the runoff water is given by Equation (9):(8)$$\varepsilon =h+\frac{aV^{2}}{2g}$$
where *a* is the kinetic energy correction coefficient (*a* = 1).

### 2.4. Measurements of Runoff, Sediment, Flow Velocity, and Rill Development

The No. 1 test plot simulated slope runoff under short-term heavy rainfall conditions, when the rainfall was 25 mm/h and the runoff washout flow was set at 2.5 m^3^/h (washout flow was calculated from the catchment area (20 m × 5 m), rainfall, and experiment water collected in the catchment area). The No. 2 plot was used as a reference to simulate the slope runoff under moderate rainfall conditions, when the rainfall was 15 mm/h and the washout flow was set at 1.5 m^3^/h. The test consisted of flushing the two test plots nine times, then measuring and recording the flow velocity, the depth and width of runoff at the 0.5, 1, 2, 3, 5, 7, 9, 11, 13, 15, 17, 19, and 20 m sections (①–⑬ are observation cross-sections), and the temperature of the runoff (Figure 2 and Figure 3). Due to the continuous undercutting of the rill and its deepening during the scouring process, the width (*W*) of the rill was determined as the width of the surface (Width 1) and the width of the rill at a 1/2 depth (Width 2), as shown in Figure 3 (Equation (9)).
(9)$$W=\frac{1}{2}\left(W\mathrm{idth}1+W\mathrm{idth}2\right)$$

The sediment content (CS) samples of runoff were collected every minute with a 500 mL beaker when runoff occurred, and the samples were then dried and weighed to calculate the sample sediment content. A total of 8 washout experiments were carried out for each plot, and 23 water samples were collected for every washout.

### 2.5. Data Analysis

A one-way ANOVA and a least significant difference (LSD) test were used with significance levels of 0.05 and 0.01 to evaluate the significant differences in the different rainfall intensities for all the factors analyzed here, using SPSS 19.0 (IBM SPSS, Inc., Chicago, IL, USA) for Windows. Origin 9.0 software (OriginLab, Northampton, MA, USA) was used to analyze the multiple linear regression with nonlinear methods, along with the relationships between sediment content and hydraulic characteristic parameters, rainfall intensity and hydraulic characteristic parameters, sediment content, and rainfall intensity. The average values and standard deviations (SD) of all factors were calculated using Microsoft Office Excel 2010 (Microsoft, Redmond, WA, USA).

## 3. Results and Analysis

### 3.1. Runoff Development

The width and depth of runoff are important manifestations of flow. The runoff width of all sections of the two plots decreased with increasing washout duration (Figure 4), while the runoff depth increased with increasing washout duration (Figure 4). The average runoff widths (*RW*_avg_) (the average value of runoff width of all measured cross-sections, washed every 25 min) of the No. 1 plot were greater than those of the No. 2 plot for all washout durations, while the *RW*^No. 1^_avg_ (*RW*_avg_ of the No. 1 plot) and *RW*^No. 2^_avg_ (*RW*_avg_ of the No. 2 plot) had a lower trend from a washout duration of 0 to 150 min, while the *RW*_avg_ showed a significant difference between the two plots and then increased after 150 min (Figure 4). The maximum *RW*^No. 1^_avg_ (80.68 cm) appeared at 25 min, while the maximum *RW*^No. 2^_avg_ (61.48 cm) appeared at 50 min. The minimum *RW*_avg_ of the two plots appeared at 150 min (No.1_min_ was 33.05 cm and No.2_min_ was 25.29 cm). At the initial stage (0–75 min) of the washout experiment, the soil surface of the plots had not yet formed rills, and the runoff was mainly in the form of plane flow. With the washout duration increasing (75–150 min), rills began to form in the two plots, and runoff gradually became a stream, which caused the *RW* of the two plots to decrease. After the formation of rills (150–200 min), the lateral erosion intensified, which caused the *RW* of the two plots to increase with increasing rill width.

The depth of runoff (*RD*) showed a different trend in terms of width; the *RD* of all sections of the two plots both increased with increasing washout duration, and *RD*^No. 1^_avg_ increased significantly, but *RD*^No. 2^_avg_ increased non-significantly (Figure 4). In the washout duration of 0 to 125 min, *RD*^No. 1^_avg_ increased with increasing washout duration, and *RD*^No. 2^_avg_ showed no significant difference with *RD*^No. 1^_avg_, while in the washout duration of 125 min to 200 min, the change in *RD*^No. 2^_avg_ with increasing washout duration was not significant, but the difference between the two plots increased gradually (Figure 4). Therefore, the maximum of *RD*^No. 1^_avg_ (6.27 cm) and *RD*^No. 2^_avg_ (5.44 cm) appeared at the washout duration of 200 min and 125 min, respectively, while the minimum values of *RD*_avg_ of the two plots (No. 1_min_ was 3.65 cm and No. 2_min_ was 3.85 cm) both appeared at a washout duration of 25 min (Figure 4).

### 3.2. Rill Development

Two rills appeared during the scouring period of 125 min. In the initial stage of the washout experiment (0–100 min), no complete rill was formed, and the runoff in the two plots was dominated by flooding, which caused the width of the rills in the two plots to be relatively wide at this stage. The RIW_avg_ (rill width) of plot No. 1 (56.56 cm) was greater than that of plot No. 2 (52.36 cm) at 100 min, but there was no significant difference between the two plots. With increasing washout duration, rills formed in the two plots at 125 min, when soil erosion decreased and the gully widths in the two plots gradually decreased, which caused the rill widths in the two plots to decrease significantly at 125 and 150 min (Figure 5).

With increasing duration, the lateral erosion of the rills in the two plots increased, and the rill width increased. After rills formed, the *RIW*^No.1^_avg_ was greater than the *RIW*^No.2^_avg_ at all washout durations, and the higher the flow was, the greater the erosion intensity. The soil layer in the test area was thin, consisting of the dry red soil in the surface layer and the hard uncultivated soil in the lower layer, which blocked the undercutting erosion and then led to the strengthening of the lateral erosion, causing the rill depth to not change significantly with increasing washout duration.

The width/depth (W/D) ratio is one of the main parameters of rill morphology, which can reflect the erosional changes in rills under different flow rates and washout durations. The results showed that the average W/D ratio of plot No. 1 (W/D^No. 1^_avg_) was greater than that of W/D^No. 2^_avg_ in all washout durations when rills formed, and both W/D ratios of the two plots showed an increasing trend with increasing washout duration. Due to the strong lateral erosion, the rills in the two plots were wide and shallow, and their W/D ratios were significantly greater than 1 (Figure 6).

### 3.3. Flow Hydrodynamic Characteristics

Soil erosion is a complex physical process involving the interactions between water flow and soil. The study of the hydraulic characteristics of runoff can deepen our understanding of the energy changes of surface runoff caused by rainfall, wherein different hydraulic parameters reflect different hydrodynamic effects. The runoff hydraulic parameters of velocity (*V*), Froude number (*Fr*), unit water flow power (*P*), cross-section energy (*ε*), and hydraulic radius (*R*) of the two plots showed the same trend with increasing washout duration. The runoff *V*_avg_ (average velocity of all sections in the plot at a certain duration)*, Fr*_avg_*, P*_avg_*, ε*_avg_, and *R*_avg_ values of the two plots showed a trend of first decreasing (25–75 min), then increasing (75–125 min), and finally stabilizing (125–200 min) with increasing washout duration (Figure 7a–e). The *V*_avg_, *P*_avg_*, ε*_avg_ and *R*_avg_ values were not significantly different between the two plots from washout durations of 25 to 100 min but were significantly different from the values for washout durations of 125 to 200 min (Figure 8a,c–e). Therefore, all maximum values of *P*_avg_ (0.122)*, ε*_avg_ (1.019 cm), and *R*_avg_ (0.122 m) of plot No. 1 occurred at a washout duration of 125 min, while the maximum values of *P*^No. 2^_avg_ (0.119) and *R*^No. 2^_avg_ (0.119 m) occurred at the washout duration’s initial stage (25 min), and the maximum value of *ε*^No. 2^_avg_ (0.819 cm) occurred at the washout duration’s last stage (Figure 8a,c–e). However, the difference in *Fr*_avg_ between the two plots was not significant over the whole washout duration. Both the *Fr*_avg_ values of the two plots reached their maximum values at the beginning of the washout test (25 min), and the maximum values of *Fr*^No. 1^_avg_ (1.45) and *Fr*^No. 2^_avg_ (1.46) were not significantly different (Figure 8b).

The changes in the Reynolds number (*Re*), Darcy-Weisbach drag coefficient (*DW-f*), and runoff shear force (*τ*) with increasing washout duration showed a different trend from the other hydraulic parameters (Figure 8f–h). The *Re*_avg_, *DW-f*_avg_, and *τ*_avg_ values of the two plots showed a trend of first increasing and then decreasing with increasing washout duration, but they showed different differences between the two plots.

The results of the test showed that the runoff in the two test plots was in a turbulent state, and the *Re* values of the No. 1 and No. 2 plots were greater than 500 (Figure 8), but the Re values of the runoff always indicated that the change of runoff from laminar to turbulent flow was gradual [69]. The *Re*_avg_ value of the No. 1 plot (*Re*^No. 1^_avg_) increased significantly between 25 and 125 min of washout duration, and the maximum value of *Re*^No. 1^_avg_ (*Re*^No. 1^_avgmax_) (3293.949) occurred at 125 min, while *Re*^No. 2^_avgmax_ (2762.723) occurred at 100 min (Figure 8f). Both the *Re*_avg_ values of the two plots decreased significantly after the maximum value occurred, and the difference between the two plots increased significantly.

When rills formed on a slope, the velocity of runoff was inevitably subjected to the resistance of the rill wall. The magnitude of the resistance affected the velocity of the runoff directly and was closely related to the effective erosion force of the runoff on the slope soil so that *DW-f* was an important index. The *DW-f*_avg_ values of the plots showed the same changing trend as the *Re*_avg_ values, while the difference between the two plots increased, but not significantly, after *DW-f*^No. 1^_avgmax_ and *DW-f*^No. 2^_avgmax_ occurred (Figure 8g).

*Τ* is generated in the direction of runoff movement. During the process of runoff movement, the effect of *τ* was to wash away and destroy the original structure of the soil, disperse the soil into particles, and transport them to the slope along with the movement of water. Both *τ*_avg_ values of the two plots fluctuated significantly with increasing washout duration, while *τ*^No. 1^_avgmax_ (5.988) and *τ*^No. 2^_avgmax_ (5.365) occurred at 125 min and 100 min, respectively (Figure 7h).

The hydraulic characteristics of runoff in the short-term heavy rainfall and moderate rainfall experimental tests showed different changing trends with increasing washout duration, and the difference in the total value between the two plots was also different. The *Re* value of the total average of all sections and all durations (*Re*_Tavg_) showed an extremely significant difference between the two plots (Figure 8), and the *Re*^No. 1^_Tavg_ value (2506.142) was significantly greater than that of the *Re*^No. 2^_Tavg_ (2099.295). The *R*_Tavg_*, τ*_Tavg,_ and *DW-f*_Tavg_ values were significantly different between the two plots, while *R*^No. 1^_Tavg_ (0.110 m)*, τ*^No. 1^_Tavg_ (70.096), and *DW-f*^No. 1^_Tavg_ (5.139) were significantly greater than *R*^No. 2^_Tavg_ (0.102 m)*, τ*^No. 2^_Tavg_ (62.918), and *DW-f*^No. 2^_Tavg_ (4.613) (Figure 8). However, *ε*_Tavg_, *P*_Tavg_*,* and *Fr*_Tavg_ were not significantly different between the two plots.

### 3.4. Sediment Content

The sediment content (CS) of flow is an important factor reflecting the process of erosion and sediment production, and the amount of sediment content directly affects the amount of erosion. Generally, under the same environmental conditions, a larger runoff flow has a greater sediment content. Short-term heavy rainfall increased the sediment content significantly (0.9–9.5%), but decreased with increasing washout duration. The results showed that CS^No. 1^_Tavg_ (0.158 g/cm^3^) was significantly different from CS^No. 2^_Tavg_ (0.153 g/cm^3^), and both CSs in the two plots decreased with increasing washout duration. In the initial stage of slope erosion, the erosion activity was violent, and the runoff had a higher CS (Figure 9). With the washout duration increasing, the changes in the depth and width of rills became steady, which caused the CS to decrease.

### 3.5. Correlations between Soil Erosion and Flow Hydrodynamics

The hydraulic characteristics of runoff are important factors for studying the mechanisms of slope soil erosion; when slope runoff reaches a certain hydraulic index, rill erosion occurs [13,23,32,68]. Rill erosion is the process of soil dispersion, separation, and transportation. Once a rill is produced, the erosion and transport forces of runoff will be much greater than that of raindrop splash and flaky water flow on the slope. To investigate the relationship between slope soil-gravel farmland soil erosion and flow hydraulics, Pearson correlation analysis was used to determine the crucial hydraulic flow parameters affecting soil loss under short-term heavy rainfall conditions (Table 1). The crucial hydrodynamic parameters of plot No. 1 that affected CS were *τ* (*r* = −0.863 **), *DW-f* (*r* = −0.863 **), and *Re* (*r* = −0.735 *) (Table 1), while the correlation coefficients between CS and *τ*, and also CS and *DW-f*, were the largest; the crucial hydrodynamic parameters of plot No. 2 that affected CS were *V* (*r* = 0.814 *), *R* (*r* = 0.811 *) and *P* (*r* = 0.811 *) (see Table 1). The relationships between sediment content (CS) and hydrodynamic flow parameters (*τ*, *DW-f*, *Re*, *V*, *R*, and *P*) were described well by the linear function (Figure 10).

## 4. Discussion

### 4.1. The Impact of Short-Term Heavy Rainfall on Rill Morphological Characteristics and Flow Hydrodynamics

#### 4.1.1. Runoff and Rill Morphological Characteristics

It is generally recognized that rills are developed mainly via concentrated overland flow and that undercutting erosion is dominant [70]. However, our results implied that short-term heavy rainfall caused the soil-gravel combined slope to be dominated by lateral erosion, supplemented by undercutting erosion. The morphological characteristics (width and depth) of runoff and the morphological characteristics of rills complemented each other. In general, the runoff width was positively related to the width and depth of the rill. In the early stage of scouring (washout duration of 0–75 min), the rills of both treatments were not formed on the slope surface when the runoff was surface flow and the depth of the runoff was also shallow, but the width was wide. With an increase in washout duration, the surface soil of the slope was eroded completely, and rills formed, which led to concentrated water flow and a decrease in the width of the runoff (washout duration of 75–150 min). The depth increased over the whole washout duration, and the width of the rills first decreased and then increased. The rill depth did not increase significantly after rill formation. In the later period of scouring (washout duration of 150–200 min), due to the intensification of lateral erosion, the width of the rills increased. At the same time, the slope runoff eroded downward and removed soil, and gravel was exposed on the surfaces, which hindered the operation of the slope water flow and reduced the average flow rate, and the lateral erosion intensified; therefore, the width of the runoff also increased. From a washout duration of 0 to 125 min, the water flow infiltrated into the soil, resulting in a smaller runoff; in the later stage of scouring, the soil was saturated with water, and the runoff completely flowed into the rills, thus, the runoff depth increased gradually with washout duration.

#### 4.1.2. Flow Hydrodynamics

The runoff velocity (*V*) was significantly reduced by the underlying surface of the soil and gravel. At the beginning of the scouring test (washout duration of 0–75 min), rills in the two plots were not formed when the flow was a surface flow, and the velocity of flow was the fastest over the whole washout duration. With the increase in washout duration, soil and gravel began to be exposed, which increased the surface roughness and caused the runoff to be greatly hindered on the gentle slope so that the velocity of flow decreased significantly. When the topsoil washed away and rills were formed, the water flow started to increase, and the soil and gravel were carried away by the runoff, causing the flow velocity to gradually increase (Figure 10a). However, flow is the main source of power for slope erosion, which determines the changes in flow velocity and hydraulic characteristics [23,68]; rainfall intensity has a significant effect on runoff flow (Table 2). Under the same environmental conditions, the higher the rainfall intensity is, the higher the flow and velocity. This study showed that the velocity of runoff had a significant correlation with rainfall intensity (0.264 **, *p* < 0.01) (Table 2), while the average velocity of runoff of short-term heavy rainfall (0.79 m/s) was greater than that of short-term rainfall (0.72 m/s), increasing by 9.7% (Figure 10a).

The velocity of flow directly affects soil separation, sediment transport, and the deposition processes of slope water erosion. It is a very important hydrodynamic parameter for studying slope soil erosion and is also a necessary parameter for calculating other hydrodynamic parameters [70]. When the velocity of the flow changed, other hydraulic characteristics also changed [15,71]. Therefore, as the most important factor affecting runoff velocity, the intensity of short-term rainfall had an extremely significant correlation with Re (*r* = 0.230 **, *p* < 0.01) and *ε* (*r* = 0.210 **, *p* < 0.01) and a significant correlation with the *R* (*r* = 0.155 *, *p* < 0.05), *P* (*r* = 0.152 *, *p* < 0.05), *τ* (0.155 *, *p* < 0.05), and *DW-f* (0.155 *, *p* < 0.05) of runoff (Table 2). Moreover, this study showed that the changing trend of runoff, in terms of *Re*, *R*, and *P* of the two plots with increasing washout duration was the same as that of the velocity. Under short-term heavy rainfall conditions, the velocity of runoff had an extremely significant correlation with *R* (*r* = 1.000 **) and *P* (*r* = 1.000 **) and a significant correlation with ***ε*** (*r* = 0.747 *), but the velocity of runoff only had an extremely significant correlation with *R* (*r* = 1.000 **) under moderate rainfall conditions. The cross-section energy (*ε*) is the sum of the kinetic and potential energies per unit weight of water in the reference surface and is the result of a range of factors, including the section shape, size, flow velocity, and water depth [72,73]. Generally, the larger the runoff velocity and the deeper the runoff is, the greater the energy per unit weight of water.

### 4.2. Flow Hydrodynamics and Sediment Content

The flow has a significant impact on soil erosion, lifting up the material and destroying the morphology of the soil surface [15,22]. When the flow is large enough, the material carried by the flow abrades the surface, and a larger flow always leads to stronger soil erosion [74,75]. As can be seen, the flow sediment content (CS) reflects the soil erosion efficiency [74]. This study showed that under short-term heavy rainfall conditions, the sediment content had an extremely significant negative correlation with *τ* (*r* = −0.863 **) and *DW-f* (*r* = −0.863 **) and a significant negative correlation with *Re* (*r* = −0.735 *) but showed no significant correlation with other hydraulic characteristics. However, under moderate rainfall conditions, the sediment content had a significant positive correlation with *V* (*r* = 0.814 *), *R* (*r* = 0.811 *), and *P* (*r* = 0.811 *), and no significant correlation with other hydraulic characteristics.

In the same catchment area, the water volume of short-term heavy rainfall was significantly larger than that of moderate rain, while a greater water volume represented greater water flow velocity. In the first half of the experiment (washout durations of 0–100 min), the difference in runoff V between the two plots was not significant. As the washout duration increased, the larger water volume caused by the short-term heavy rainfall removed the soil and most of the gravel; only part of the large gravel remained (Figure 11), and the resistance was significantly reduced. Moderate rain only removed the soil and part of the small gravel in the rills, due to the small amount of water and flow velocity, while the gravel provided greater resistance to the water flow (Figure 11). Therefore, in the second half of scouring (washout durations of 100–200 min), the runoff velocity formed by short-term heavy rainfall was significantly higher than that under moderate rainfall conditions. At the same time, due to the reduction in resistance, the runoff energy of short-term heavy rainfall was used more for rill wall soil, and it stabilized the water flow with the increase in sand content, resulting in a decrease in *Re* and *τ*.

Usually, the runoff resistance’s changing trend was consistent with the runoff, *DW-f*, and the greater the runoff resistance was [66,76], the greater the energy consumed by runoff to overcome the slope resistance, and the lesser the energy used for slope erosion and sediment transport, which caused the sediment content to have a negative correlation with *DW-f*. Similarly, *τ* is an important factor affecting runoff sediment content and is one of the hydrodynamic parameters most closely related to the changing process of sediment content in flow under test conditions; the greater *τ* is, the lower the runoff sediment content, and vice versa. *Re* is a measure of the ratio of the inertial force to the viscous force of a fluid, and it is a dimensionless number. When *Re* is small, the influence of the viscous force on the flow field is greater than the inertial force, and the disturbance of the flow velocity in the flow field is attenuated by the viscous force, with stable and laminar fluid flow. In contrast, if *Re* is large, the inertial force on the flow field is greater than the viscous force, with a relatively unstable fluid flow; a slight change in the flow velocity easily develops and strengthens, forming a turbulent and irregular turbulent flow field [71,77]. Moreover, with increasing sediment content, the stability of runoff increases; thus, the *Re* of the flow decreases with increasing sediment content [62,78]. Therefore, the sediment content had an extremely significant negative correlation with *τ* and *DW-f* and a significant negative correlation with *Re*.

Under moderate rain conditions, in the second half of the washout duration (100–200 min), the water flow was resisted greatly, and the *V* of the runoff decreased significantly. However, the erosion mode was dominated by undercutting erosion around the gravel, and the sediment content of runoff did not decrease but instead increased, which led to the *V* (*r* = 0.814 *) of runoff having a significant positive correlation with CS. In addition, the *R* and *P* of runoff always had an extremely significant positive correlation with *V*, and the *R* and *P* of runoff changed with changes in *V*. This study showed that the *R* (*r* = 1.000 **) and *P* (*r* = 1.000 **) of both plots had an extremely significant correlation with *V*, which caused the *R* (*r* = 0.811 *) and *P* (*r* = 0.811 *) of the runoff to have a significant positive correlation with CS.

Conversely, there was a significant correlation between soil erosion and environmental quality, even public health [15]. The influence of soil erosion on soil productivity is significant, leading to the loss of the cultivated layer, the decline of soil fertility, soil organic matter, and effective plant water storage, along with an increase in soil thermal reflectance, soil crusting, and also soil compaction. At the same time, due to the input of dissolved and particle-bound nutrients, this increases the nutrients in the water receiving runoff and decreases the water quality, accompanied by the risk of an increase in eutrophication [38,39]. Especially in agricultural soil with high nutrient content, the runoff and erosion sediment contain rich nutrients and also usually carry a high nutrient load [9,59,60]. In the past 10 years, the green vegetable industry in the Yuanmou dry-hot valley has developed rapidly. The vegetable planting area has reached more than 16,000 hectare, of which more than 30% is slope land [61,62]. After a long dry season, the loose soil surface in Yuanmou dry-hot valley would be eroded in the rainy season very easily. Therefore, it will inevitably become a potential source of water environmental pollution. Meanwhile, more rainfall means a stronger scouring force, leading to more soil erosion, and more pollution sources will be carried to rivers and lakes [9,59,60]. These processes often accompany and affect each other, resulting in serious ecological and environmental problems.

In China, water and soil loss is also an important reason for the decline in water quality in rivers and lakes, with 5.6% being oligotrophic lakes, 44.4% mesotrophic lakes, and 50% eutrophic lakes. With no exception in terms of the lakes in the middle and lower reaches of the Yangtze River, more than half of the lakes have been affected by eutrophication to varying degrees, especially those in the suburbs. There are many erosion sources that can cause water pollution, but eroded soil is the ultimate source of sediment, and a higher CS always results in more serious ecological and environmental problems in rivers and lakes. Therefore, in the Yuanmou dry-hot valley, short-term heavy rainfall caused a higher sediment content than with moderate rainfall, which would cause the ecological and environmental problems in rivers and lakes to be more serious.

## 5. Conclusions

A field study focusing on the impacts of short-term heavy rainfall and moderate rainfall on soil erosion on gentle soil-gravel slopes was conducted to explore the effect of the dynamic changes of hydraulic characteristics and sediment content on rill formation, as well as the interactions between hydraulic characteristics and sediment content during soil erosion. Compared to moderate rainfall, short-term heavy rainfall increased the sediment content, which was mainly due to the velocity and hydraulic characteristics of runoff when impacted by flow and soil-gravel conditions. Short-term heavy rainfall significantly strengthened rill evolution under soil-gravel conditions, which mainly resulted in increases in rill width and runoff velocity, thereby affecting the hydraulic characteristics of the runoff. Short-term heavy rainfall significantly increased the sediment content (0.9–9.5%), but this increase decreased with increasing washout duration. Rainfall and soil-gravel conditions played important roles in soil loss; the most sensitive hydrodynamic parameters to soil loss were the Darcy-Weisbach drag coefficient, Reynolds number, and runoff shear force under short-term heavy rainfall conditions, which had a significant negative correlation with sediment content. The most sensitive hydrodynamic parameters to soil loss were velocity, hydraulic radius, and unit stream power under moderate conditions; they had a significant positive correlation with sediment content. Higher CS always results in more serious ecological and environmental problems in rivers and lakes; in this study, short-term heavy rainfall caused higher sediment content than with moderate rainfall, causing the ecological and environmental problems of rivers and lakes to be more serious.

These findings will help us to understand the impacts of short-term heavy rainfall on rill development, the hydrodynamic mechanisms of soil erosion, and the interactions between erosion processes on gentle soil-gravel slopes. Weakening the effects of the flow of short-term heavy rainfall impacts, such as taking measures to divert the flow, is useful for erosion control and the prevention of rill formation. In addition, consideration of the interactions between soil erosion processes that are driven by different rainfall flows could lead to improvements in process-based erosion models. Research is needed to better understand the dynamic interactions between rill development and different rainfall and gravel amounts on gentle slopes. On the one hand, the soil for this test is a single type and the test slope is consistent, but on the other hand, the local differences in soil properties and the inconsistency of regional topography have not been sufficiently analyzed. Meanwhile, the value calculated according to the average value only reflects the average condition of the slope, ignoring the uneven distribution of runoff erosion capacity, which was also proposed by previous scholars. Further research is needed to lay a foundation for exploring the influencing factors of the hydraulic parameter’s changing characteristics in the future.

## Figures and Tables

**Figure 1 ijerph-19-15232-f001:**
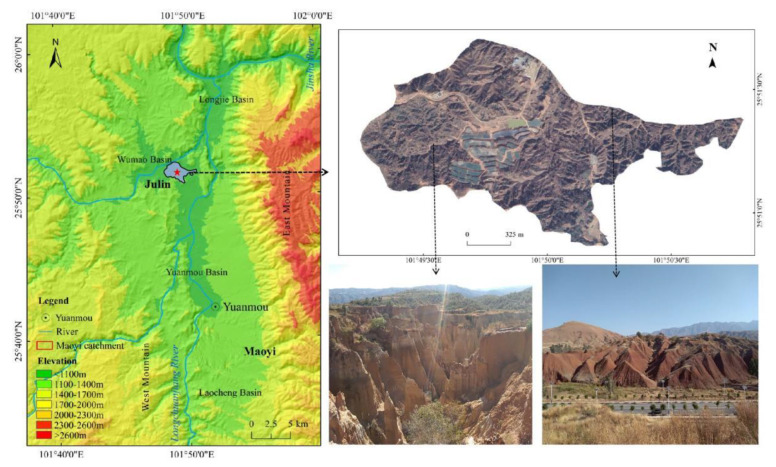
Location and landscape background of the study area.

**Figure 2 ijerph-19-15232-f002:**
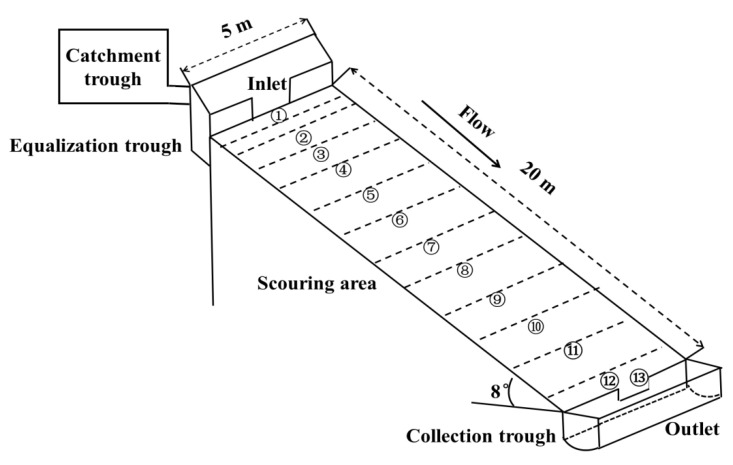
Sketch of the simulated scouring experiment with two plots (note that ①–⑬ are the observation cross-sections).

**Figure 3 ijerph-19-15232-f003:**
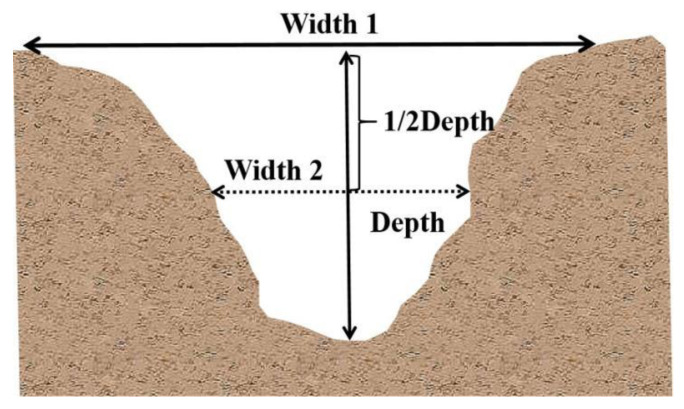
Schematic diagram of rill morphological parameters.

**Figure 4 ijerph-19-15232-f004:**
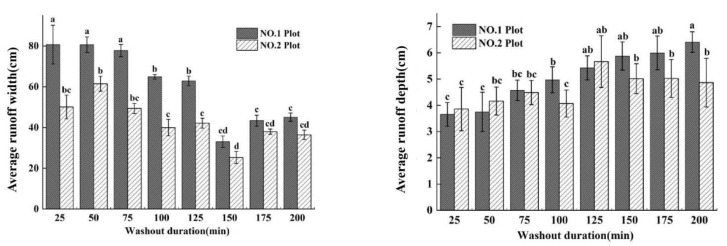
The changes in average runoff width and depth of short-term heavy rainfall (No. 1 plot) and moderate rainfall (No. 2 plot), with washout duration increasing. Note: The same letter on the histogram indicates no significant difference, and different letters indicate significant differences (*p* < 0.05). The same applies below.

**Figure 5 ijerph-19-15232-f005:**
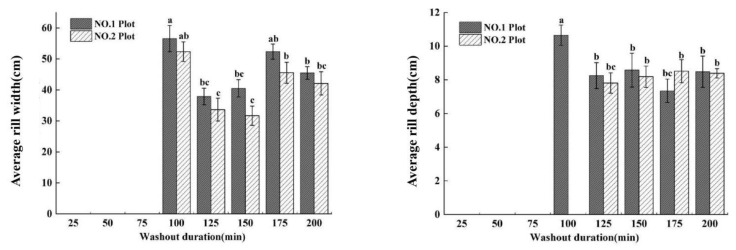
The changes in average rill width and depth of short-term heavy rainfall (No. 1 plot) and moderate rainfall (No. 2 plot) with washout duration increasing.

**Figure 6 ijerph-19-15232-f006:**
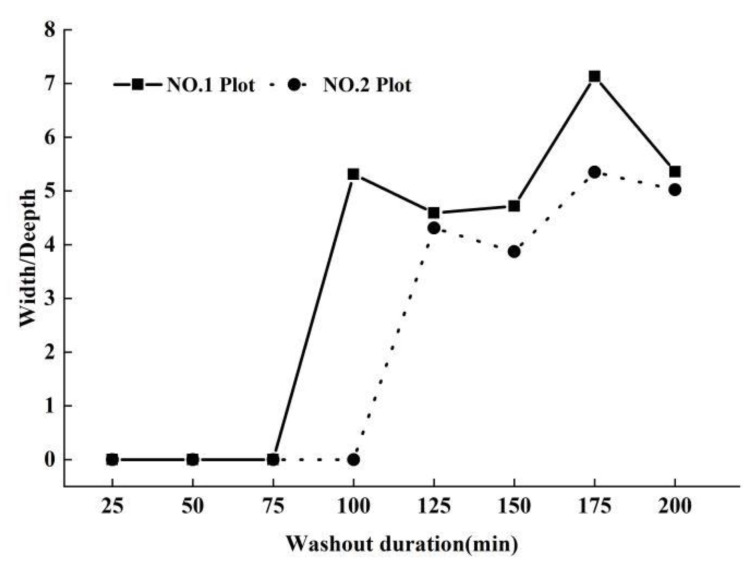
The dynamic changes in rill width/depth (W/D) with washout duration increasing, in short-term heavy rainfall (No. 1 plot) and moderate rainfall (No. 2 plot) experiments.

**Figure 7 ijerph-19-15232-f007:**
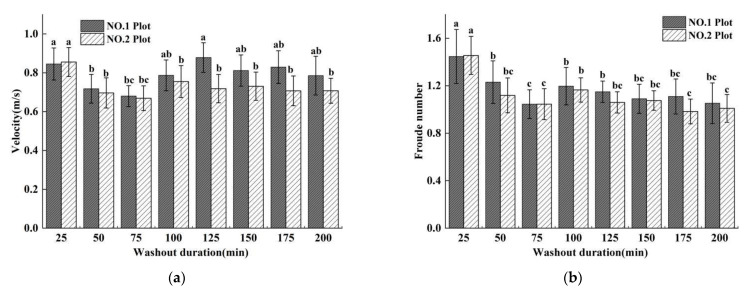
Average velocity (*V*) (**a**), Froude number (*Fr*) (**b**), Unit stream power (*P*) (**c**), Cross-section energy(*ε*) (**d**), Hydraulic radius (*R*) (**e**), Reynolds number (*Re*) (**f**), Darcy-Weisbach drag (*DW-f*) (**g**), and Runoff shear force (*τ*) (**h**) in short-term heavy rainfall (No. 1 plot) and moderate rainfall (No. 2 plot) experiments.

**Figure 8 ijerph-19-15232-f008:**
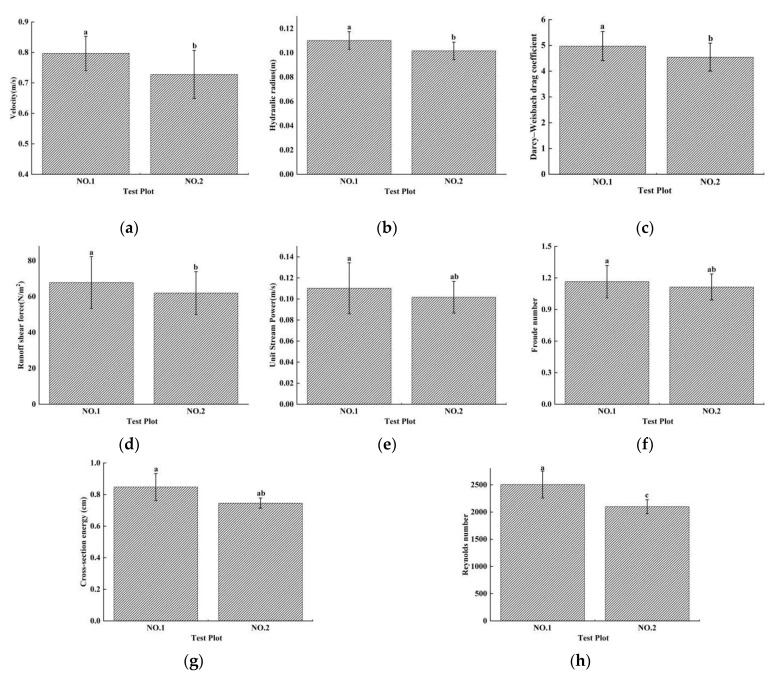
Total velocity (*V*) (**a**), Hydraulic radius (*R*) (**b**), Darcy-Weisbach drag (*DW-f*) (**c**), Runoff shear force (*τ*) (**d**), Unit stream power (*P*) (**e**), Froude number (*Fr*) (**f**), Cross-section energy (*ε*) (**g**) and Reynolds number (*Re*) (**h**) in short-term heavy rainfall (No. 1 plot) and moderate rainfall (No. 2 plot) experiments.

**Figure 9 ijerph-19-15232-f009:**
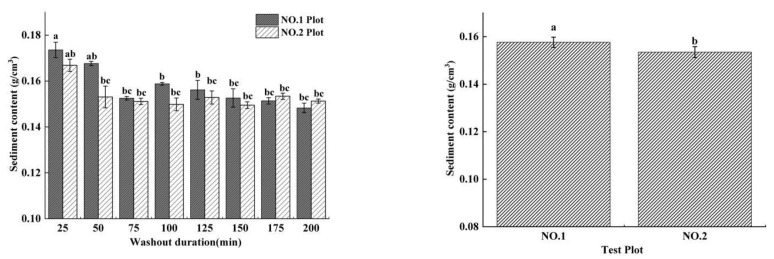
Average and total sediment content of short-term rainfall and moderate rainfall experiments.

**Figure 10 ijerph-19-15232-f010:**
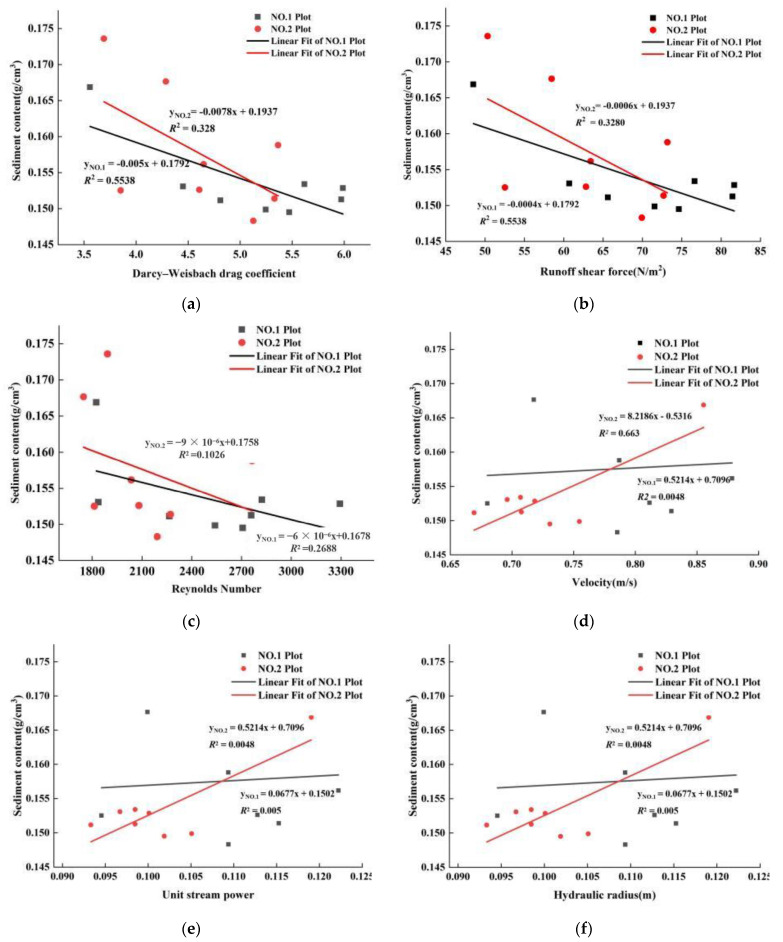
The relationship between mean sediment content (CS) and mean Darcy-Weisbach drag (*DW-f*) (**a**), Runoff shear force (*τ*) (**b**), Reynolds number (*Re*) (**c**), Velocity (*V*) (**d**), Unit stream power (*P*) (**e**) and Hydraulic radius (*R*) (**f**) in short-term heavy rainfall and moderate rainfall experiments.

**Figure 11 ijerph-19-15232-f011:**
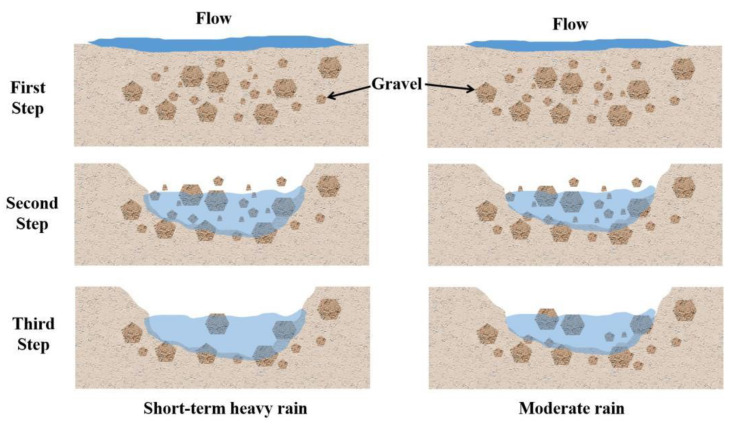
Schematic diagram of the washout process of short-term heavy rainfall and moderate rainfall.

**Table 1 ijerph-19-15232-t001:** Correlation between the mean sediment content and runoff hydraulic parameters in the two experiments.

Parameters	CS (g/cm^3^)	*V* (m/s)	*Re*	*R* (m)	*Fr*	*τ* (N/m^2^)	*P* (m/s)	*ε* (cm)
No. 1 plot (16.67 mm/min)								
*V*	0.069							
*Re*	−0.735 *	0.525						
*R*	0.071	1.000 **	0.525					
*Fr*	0.183	−0.091	−0.328	−0.091				
*τ*	−0.863 **	0.248	0.921 **	0.249	−0.432			
*P*	0.071	1.000 **	0.525	1.000 **	−0.091	0.249		
*ε*	0.599	0.747 *	0.912 **	0.747 *	−0.254	0.801 *	0.747 *	
*DW-f*	−0.863 **	0.248	0.921 **	0.249	−0.432	1.000 **	0.249	0.801 *
No. 2 plot (10 mm/min)								
*V*	0.814 *							
*Re*	0.335	0.404						
*R*	0.811 *	1.000 **	0.410					
*Fr*	0.309	0.052	0.042	0.052				
*τ*	−0.571	−0.269	0.040	−0.268	−0.507			
*P*	0.811 *	0.410	0.410	1.000 **	0.052	−0.268		
*ε*	0.164	0.200	0.756 *	0.204	−0.294	0.390	0.204	
*DW-f*	−0.571	−0.269	0.040	−0.268	−0.507	1.000 **	0.390	−0.573

Note: ** indicates the correlation is extremely significant, * indicates the correlation is significant.

**Table 2 ijerph-19-15232-t002:** The effect of different flows on hydraulic parameters.

Parameters	*V*	*R*	*Re*	*Fr*	*DW-f*	*τ*	*P*	*ε*
*r*	0.264 **	0.155 *	0.230 **	0.039	0.155 *	0.155 *	0.152 *	0.210 **
*p*	0.001	0.017	0.001	0.548	0.017	0.017	0.02	0.001

Note: ** indicates the correlation is extremely significant, * indicates the correlation is significant.

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
