# Peer review of "The Influence of Short-Term Heavy Rainfall on Hydraulic Characteristics and Rill Formation in the Yuanmou Dry-Hot Valley"

_ijerph, 2022, doi:10.3390/ijerph192215232_

Round 1
Reviewer 1 Report
In the dry-hot valley of Jinsha River, rainfall is concentrated, soil erosion resistance is weak, and the surface is exposed, which leads to serious soil erosion and brings great adverse effects to regional ecological construction and agricultural production. Therefore, the study on the development process and influencing factors of rill erosion on the slope in dry-hot valley can provide scientific basis for the prevention and control of soil erosion on the slope. In general, the test data of this study is detailed and the content analysis is comprehensive, and the research on the influence of hydrodynamic process on the development process of slope rill erosion in dry-hot valley area has certain innovative significance.
However, there are still some problems in the paper, which need to be further modified and improved.
1. In the abstract of the paper, it is suggested that the author refine the research significance according to the characteristics of dry and hot valleys, because there have been many studies on rill erosion in other regions.
2. There may be some problems in the experimental design of this study. By designing runoff processes under different rainfall intensities, the authors analyzed the impact of runoff on rill erosion development. However, from the test design of the paper, it is just the rill erosion process formed by the upper water on the slope, which is more different from the rill erosion phenomenon produced by concentrated streams formed by the convergence of rainfall process on the slope. It is suggested that the author make further analysis and improvement.
3. In fact, the rill erosion process is closely related to not only the hydrodynamic characteristics, but also the soil characteristics. However, the physical and chemical properties of the test soil samples, such as bulk density, mechanical composition, infiltration characteristics and other soil parameters, are basically not introduced in this paper. It is suggested that the author further analyze the soil treatment of the two plots to ensure that the two plots are consistent in soil treatment.
4. From the test photos, human trampling and other disturbance factors may have a greater impact on the test process, resulting in different levels of soil treatment in the two plots. It is suggested that the author consider this problem, otherwise it will lead to large deviation to the research results.
5. It is suggested that the author introduce the determination method of slope hydrodynamic parameters in Section 2.3.
6. In view of the possible shortcomings in the paper, it is suggested to further analyze the areas that need to be improved in the future in the discussion section.
7. There are still some details in the paper writing. For example, the abstract does not explain what "DW" means. Similar problems need further modification and improvement.
Author Response
Dear Editors and Reviewers:
Thank you for your letter and for the reviewers’ comments concerning our manuscript entitled “The influence of short-term heavy rainfall on hydraulic characteristics and rill formation in the Yuanmou dry-hot valley” (ijerph-1994563). Those comments are very valuable and very helpful for revising and improving our paper, as well as the important guiding significance to our researches. We have studied comments carefully and have made correction which we hope meet with approval. Revised portion are marked in red in the paper. The main corrections in the paper and the responds to the reviewer’s comments are as flowing.
- In the abstract of the paper, it is suggested that the author refine the research significance according to the characteristics of dry and hot valleys, because there have been many studies on rill erosion in other regions.
Respond:We rewrite the abstract and refine the research significance according to the characteristics of dry and hot valleys.
ABSTRACT: Rill erosion is one of the major environmental problems in the world, which is an important factor in regard to land degradation and has a serious impact on production and life in the region. The widely distributed Yuanmou group stratum promotes the development of rill erosion, where the strong time-concentrated rainfall and the alternating arid-humid climate prepare the ground for the development of rill in soils. Therefore, the study of the processes of slope rill erosion was carried out, and a gravel–soil slope in the Yuanmou dry-hot valley was chosen to simulate short-term heavy rainfall (25 mm/h) (No. 1 plot) and moderate rainfall (15 mm/h) (No. 2 plot) to study the erosion processes of soil and the dynamic characteristics of runoff involved in erosion. The study results showed that the width of runoff was significantly different between the two plots, while the depth of runoff was not significantly different. When rills formed, the width of the two plots first decreased and then increased with increasing washout duration, while its depth did not change significantly. Flow was the key factor in determining the hydraulic characteristics of runoff, and it had a significant or extremely significant positive correlation with hydraulic characteristic parameters, except for Fr(Froude number) (r = 0.039)., respectively. The total sediment content (CS) of plot No. 1 (0.158 g/cm3) was significantly different from that of plot No. 2 (0.153 g/cm3), and both CSs in the two plots decreased with increasing washout duration. The CS had an extremely significant negative correlation with τ(Runoff shear force) (r = -0.863**) and DW-f(Darcy–Weisbach drag coefficient) (r = -0.863**) and a significant negative correlation with Re(Reynolds number) (r = -0.735*) in the short-term heavy rainfall experiment, and the CS had a significant negative correlation with V(Velocity) (r = 0.814*), R(Hydraulic radius) (r = 0.811*) and P(Unit stream power) (r = 0.811*) in the moderate rainfall experiment. The results of this study will help guide further examination of the processes involved in and the dynamic mechanisms of rill erosion on slopes under short-term heavy rainfall conditions.
- There may be some problems in the experimental design of this study. By designing runoff processes under different rainfall intensities, the authors analyzed the impact of runoff on rill erosion development. However, from the test design of the paper, it is just the rill erosion process formed by the upper water on the slope, which is more different from the rill erosion phenomenon produced by concentrated streams formed by the convergence of rainfall process on the slope. It is suggested that the author make further analysis and improvement.
Respond:Thanks for your advise.The Precipitation of Yuanmou dry hot valley during the rainy season (May-October) accounts for 91% of the annual rainfall. The rainfall pattern of Yuanmou dry-hot valleys mainly shows short term heavy rainfall. At the same time, the soil types in the valley include dry red soil and vertisol, the soil water storage capacity is poor. Therefore, the short-term heavy rainfall in Yuanmou dry-hot valley leads to the rapid convergence of the ground laminar flow to form the stock flow. Based on this, the experimental design is carried out in this paper. Further analysis and improvement have been made in accordance with the requirements of reviewers. Please check the revised manuscript for details.
- In fact, the rill erosion process is closely related to not only the hydrodynamic characteristics, but also the soil characteristics. However, the physical and chemical properties of the test soil samples, such as bulk density, mechanical composition, infiltration characteristics and other soil parameters, are basically not introduced in this paper. It is suggested that the author further analyze the soil treatment of the two plots to ensure that the two plots are consistent in soil treatment.
Respond:The soil of the plots was divided into upper and lower layers. The upper layer consisted of the typical dry red soil of the local area, and samples of this soil were collected within 20 m of the test plot. The lower layer consisted of vertisol, and soil samples were collected from nearby areas (<1 km). The depth of both soil layers was 0.5 m, and the plot materials, such as stones or clods, were not cleaned and retained their original soil–gravel composition (Fig. 3). However, the plants in the test plot were removed before the test and the slope was flattened. A waiting period of approximately 30 days then followed so that the test slope was close to the condition of a real slope. Dry red soil is also known as Jianyu dry-moist iron rich soil in terms of systematic classification. Due to weak weathering and eluviation, dry red soil belongs to sandy soil, which is mainly characterized by high sand content and good permeability. Vertisol has strong swelling, shrinking and disturbance characteristics, which belongs to soil with high clay content and poor permeability. Compared with dry red soil, vertisol has higher pH value and cation exchange capacity.
- From the test photos, human trampling and other disturbance factors may have a greater impact on the test process, resulting in different levels of soil treatment in the two plots. It is suggested that the author consider this problem, otherwise it will lead to large deviation to the research results.
Respond:Thanks for your advise. The soil was compacted for 30 days in the plot experiment, and the soil was close to the condition of a real slope, followed by the scouring experiment. During the experiment, due to the limitation of measuring instruments, we could not avoid entering the quadrats during the measurement. However, we reduced the impact of human factors on the scour experiment by reducing personnel entering the plots and measuring the hydraulic parameters of the sections at fixed positions. The original figure 3 has been uploaded as supplementary material.
- It is suggested that the author introduce the determination method of slope hydrodynamic parameters in Section 2.3.
Respond: we introduce the determination method of slope hydrodynamic parameters in Section 2.3.
- In view of the possible shortcomings in the paper, it is suggested to further analyze the areas that need to be improved in the future in the discussion section.
Respond: We rewrite the the discussion section, and further analyze the areas that need to be improved in the future. The soil for this test is single, and the test slope is consistent, but the local difference of soil properties and the inconsistency of regional topography has not been sufficiently analysed. Meanwhile, the value calculated according to the average value only reflects the average condition of the slope, ignoring the uneven distribution of runoff erosion capacity, which was also proposed by previous scholars . Further research is needed to lay a foundation for exploring the influencing factors of hydraulic parameter change characteristics in the future.
- There are still some details in the paper writing. For example, the abstract does not explain what "DW" means. Similar problems need further modification and improvement.
Respond: we explain "DW-f" means, and the further modification and improvement had been made in similar problems. Please check the revised manuscript for details.

Reviewer 2 Report
In this paper, the authors examined the influence of short-term heavy rainfall on hydraulic characteristics and rill formation in the Yuanmou dry-hot valley. The soil erosional process and dynamic of runoff in stimulated short-term heavy rain was examined and compared to moderate rainfall. Overall, the paper is well-written, and the conclusion is solid. My major concern is that the topic needs to fit better with the aim and scope of this journal, where the majority of the papers focus on the environmental health sciences and public health. The paper would be acceptable if the authors included the dynamic of certain pollutants, e.g., nitrogen or heavy metals, associated with the simulated rainfalls and discuss the concern of environmental quality or public health in their research.
Other comments: some figures need to be improved.
- Move Figure 3 to supplementary.
- Combine Figure 5 and Figure 6. The figure caption of figure 5 should not be on a different page from the figure.
For Figures 5, 6, 7, and 8, two sub-figures of each figure show the same data in two ways. The authors may want to make the results transparent, but they seem reductant. Besides, the authors use bar plots and averaged values while the normality of the data is questionable. In the methods, the authors should give details about how the significance test was conducted for the statistical letters of each bar.
Author Response
Dear Editors and Reviewers:
Thank you for your letter and for the reviewers’ comments concerning our manuscript entitled “The influence of short-term heavy rainfall on hydraulic characteristics and rill formation in the Yuanmou dry-hot valley” (ijerph-1994563). Those comments are very valuable and very helpful for revising and improving our paper, as well as the important guiding significance to our researches. We have studied comments carefully and have made correction which we hope meet with approval. Revised portion are marked in red in the paper. The main corrections in the paper and the responds to the reviewer’s comments are as flowing.
- In this paper, the authors examined the influence of short-term heavy rainfall on hydraulic characteristics and rill formation in the Yuanmou dry-hot valley. The soil erosional process and dynamic of runoff in stimulated short-term heavy rain was examined and compared to moderate rainfall. Overall, the paper is well-written, and the conclusion is solid. My major concern is that the topic needs to fit better with the aim and scope of this journal, where the majority of the papers focus on the environmental health sciences and public health. The paper would be acceptable if the authors included the dynamic of certain pollutants, e.g., nitrogen or heavy metals, associated with the simulated rainfalls and discuss the concern of environmental quality or public health in their research.
Respond:Thanks for your advise. Due to the loss of water samples and evaporation of water, nitrogen or heavy metals cannot be accurately measured. Therefore, this paper discuss the concern of environmental quality or public health by analyzing the sources of agricultural soil erosion and water quality reduction in rivers and lakes.
On the other hand, there has a significant correlation between soil erosion and environmental quality, even public health (Xiao and Yao et al., 2017). The influence of soil erosion on soil productivity is significant, leading to the loss of cultivated layer, the decline of soil fertility, soil organic matter and plant effective water storage, the increase of soil thermal reflectance, the soil crusting and also the soil compaction. At the same time, due to the input of dissolved and particle bound nutrients, it increases the nutrients in the water receiving runoff and decreases the water quality, accompanied by the risk increases of eutrophication (Wang and Fan et al., 2017; Onet and Dincă et al., 2019). Especially in the agricultural soil with high nutrient content, the runoff and erosion sediment contain rich nutrients and also usually high nutrient load (Ludwig and Boiffin et al., 1995; Govers and Giménez et al., 2007; Wang and Zheng et al., 2013). In the past 10 years, the green vegetable industry in Yuanmou dry hot valley has developed rapidly. The vegetable planting area has reached more than 250000 mu, of which more than 30% is slope land (Lin and Pastor et al., 2019; Sun and Chen et al., 2019). After a long dry season, Yuanmou dry hot valley in the rainy season, and the loose soil surface is very easy to be eroded. Therefore, it will inevitably become a potential source of water environment pollution. Meanwhile, more rainfall means stronger scouring force, leading to more soil erosion and more pollution sources were be carried to rivers and lakes (Ludwig and Boiffin et al., 1995; Govers and Giménez et al., 2007; Wang and Zheng et al., 2013). These processes often accompany and affect each other, resulting in serious ecological and environmental problems.
In China, water and soil loss is also an important reason for the decline of water quality in rivers and lakes, with 5.6% oligotrophic lakes, 44.4% mesotrophic lakes and 50% eutrophic lakes. With no exception of lakes in the middle and lower reaches of the Yangtze River, more than half of lakes have been affected by eutrophication to varying degrees, especially those in the suburbs. Other erosion sources are also important reasons for these water pollution, but eroded soil is the ultimate source of sediment, and higher CS always means more serious ecological and environmental problems in rivers and lakes. Therefore, in Yuanmou dry-hot valley, short-term heavy rainfall caused the higher sediment content than moderate rainfall, and that would cause the ecological and environmental problems of rivers and lakes more seriously.
- Move Figure 3 to supplementary.
Respond:We move Figure 3 to supplementary.
3.Combine Figure 5 and Figure 6. The figure caption of figure 5 should not be on a different page from the figure.
Respond:Thanks for your advise. I have revised it in the paper, please check the revised manuscript for details.
- For Figures 5, 6, 7, and 8, two sub-figures of each figure show the same data in two ways. The authors may want to make the results transparent, but they seem reductant. Besides, the authors use bar plots and averaged values while the normality of the data is questionable. In the methods, the authors should give details about how the significance test was conducted for the statistical letters of each bar.
Respond: Thanks for your advise. One of sub-figures of Figures 5, 6, 7, and 8 in the original paper was deleted, and recombined the figures into 4 and 5.
In the methods, we give the details about how the significance test was conducted for the statistical letters of each bar.
2.5. Data analysis
One-way ANOVA and least significant difference(LSD)test were used with significance levels of 0.05 and 0.01 to evaluate the significant differences in different rainfall intensity for all of the factors analyzed here using SPSS 19.0 (IBM SPSS, Inc., Chicago, IL, U.S.A.) for Windows. The multiple linear regression and nonlinear methods, the relationships between sediment content and hydraulic characteristic parameters, rainfall intensity and hydraulic characteristic parameters, sediment content and rainfall intensity were analyzed by Origin 9.0 software. Average values and standard deviations (SD) of all factors were calculated by Microsoft Office Excel 2010.

Round 2
Reviewer 1 Report
The author has made a comprehensive revision on the basis of the previous draft, which basically meets the requirements for publishing the paper. However, there are still some problems in details. The following suggestions are for the author's reference in further revision:
1. Check whether the format of full text writing, pictures, and references are consistent with the requirements of the publication.
2. It is recommended to comprehensively check the details of other aspects of the paper to ensure that the full text description is more accurate and concise.
Author Response
- Check whether the format of full text writing, pictures, and references are consistent with the requirements of the publication.
Respond:Thanks for your advise. We check the format of full text writing, pictures, and references, and please check the details from manuscript.
- It is recommended to comprehensively check the details of other aspects of the paper to ensure that the full text description is more accurate and concise.
Respond:Thanks for your advise. We check the abbreviations, tables, figures,reference, word spelling and the expression and context logic of the paper, and modified. Please check the details from manuscript.
